# Expression characteristics and interaction networks of microRNAs in spleen tissues of grass carp (*Ctenopharyngodon idella*)

**Yinli Zhao[1], Shengxin Fan[2], Pengtao Yuan[2], Guoxi Li[2]***

**1** College of Biological Engineering, Henan University of Technology, Zhengzhou, Henan Province, P.R. China, **2** College of Animal Science and Technology, Henan Agricultural University, Zhengzhou, Henan Province, P.R. China

* liguoxi0914@126.com

**Data Availability Statement:** The raw sequencing data had been deposited into the NCBI database Sequence Read Archive under number

## Abstract

The spleen is an important immune organ in fish. MicroRNAs (miRNAs) have been shown to play an important role in the regulation of immune function. However, miRNA expression profiles and their interaction networks associated with the postnatal late development of spleen tissue are still poorly understood in fish. The grass carp (*Ctenopharyngodon idella*) is an important economic aquaculture species in China. Here, two small RNA libraries were constructed from the spleen tissue of healthy grass carp at one-year-old and three-year-old. A total of 324 known conserved miRNAs and 9 novel miRNAs were identified by using bioinformatic analysis. Family analysis showed that 23 families such as let-7, mir-1, mir-10, mir-124, mir-8, mir-7, mir-9, and mir-153 were highly conserved between vertebrates and invertebrates. In addition, 14 families such as mir-459, mir-430, mir-462, mir-7147, mir-2187, and mir-722 were present only in fish. Expression analysis showed that the expression patterns of miRNAs in the spleen of one-year-old and three-year-old grass carp were highly consistent, and the percentage of miRNAs with TPM > 100 was above 39%. Twenty significant differentially expressed (SDE) miRNAs were identified. Gene ontology (GO) and the Kyoto Encyclopedia of Genes and Genomes (KEGG) pathway enrichment analysis showed that these SDE miRNAs were primarily involved in erythrocyte differentiation, lymphoid organ development, immune response, lipid metabolic process, the B cell receptor signaling pathway, the T cell receptor signaling pathway, and the PPAR signaling pathway. In addition, the following miRNA-mRNA interaction networks were constructed: immune and hematopoietic, cell proliferation and differentiation, and lipid metabolism. This study determined the miRNA transcriptome as well as miRNA-mRNA interaction networks in normal spleen tissue during the late development stages of grass carp. The results expand the number of known miRNAs in grass carp and are a valuable resource for better understanding the molecular biology of the spleen development in grass carp.

PRJNA328412 (mRNA) and PRJNA645243 (miRNA). The other data and material used and analyzed in this study are within the paper and its Supporting Information files.

**Funding:** This work was supported by grants from the Project of Henan University of Technology (CN) (No.2017XTCX04).

**Competing interests:** The authors have declared that no competing interests exist

## Introduction

The spleen is an exclusively vertebrate organ and has important functions in immunity and hematopoiesis [1]. Similar to mammals, the spleen of fish also stores and produces erythrocytes, removes the aged and affected red blood cells [2], captures and destroys blood-borne pathogens [3, 4], and induces the adaptive immune response [5, 6]. Therefore, elucidating the regulatory mechanism of spleen function may assist in understanding the immune system thereby controlling infectious diseases in fish. It is known that the spleen's basic histological architecture and related roles have been conserved during evolution throughout vertebrate species from fish to mammals [7]. Previous studies have shown that the development of the spleen begins with the condensation of the mesodermal mesenchyme underlying the dorsal mesogastrium epithelium. Next, there is the formation of a distinct splenic primordium on the dorsal stomach. The final organization of the spleen is established with the formation of the white pulp and the red pulp [8]. Despite differences in the time nodes of spleen embryogenesis, as well as the late stage development among vertebrates, the development of the spleen involves many biological processes including the growth of vascular channels, the influx of reticular cells to form a filtering network, migration of cells from other organs, and final maturation of transient and resident cell populations [9]. With the outbreak of infectious diseases in aquaculture environments, the research around the spleen's physiological function in fish has been increasingly valued [10–12]. However, understanding of fish spleen physiology is poor at the molecular level and is still largely restricted to a few species such as zebrafish [10], catfish [11], and sea bass [12]. Therefore, it is necessary to elucidate underlying molecular regulatory mechanisms of the spleen's physiological function in fish.

MicroRNAs (miRNAs) are endogenous, small, non-coding RNAs approximately 22 nucleotides in length. They have an essential function in gene regulation by binding to the 3′-untranslated region (UTR) of target mRNAs, leading to mRNA degradation or the inhibition of mRNA translation [13]. Accumulating evidence showed that miRNAs participate in the control of many biological processes, such as cell proliferation and differentiation, growth, development, organogenesis, immunity, and metabolism [14–16]. In particular, some studies have demonstrated that miRNAs are also involved in the regulation of spleen function in fish. For instance, 509 known miRNAs and 1157 novel miRNAs have been identified from four tissues of *Micropterus salmoides*, including the spleen [17]. In addition, 116 miRNAs from grouper spleen-derived cells [18], 194 conserved miRNAs and 12 novel miRNAs from the common carp spleen [19], and 452 miRNAs from *C. semilaevis* immune tissues (liver, head kidney, spleen, and intestine) [20] were also identified separately. It has been demonstrated that miR-182-3p was up-regulated after *Vibrio parahaemolyticus* flagellin stimulation in grouper spleen cells, and could directly target *TLR5M* [21]. In the spleen of Japanese flounder infected by *Edwardsiella tarda*, miR-194a was upregulated to a significant extent suppressing the type I interferon response [22]. These studies indicate that miRNA plays an important role in the regulation of spleen function and also provide new insight into the molecular biology of the spleen's physiological function in fish. However, these studies largely focused on the role of miRNAs in the spleen upon exposure to antigens and chemicals, as well as other stress conditions. At present, research on miRNA expression and function in normal spleen tissues is relatively lacking.

The grass carp (*Ctenopharyngodon idella*) is an important economic species for freshwater aquaculture in China. The recent growth in the grass carp aquaculture industry is accompanied by increasingly severe infections induced by viruses, bacteria and parasites [23]. Therefore, the prevention and treatment of infectious diseases have become the most prominent problem in grass carp aquaculture. Considering the economic importance of this fish species,

the research regarding the immune system and its function has received considerable attention. Recent studies have focused on miRNAs in the regulation of grass carp spleen function. For example, a total of 160 conserved miRNAs and 18 novel miRNAs were identified from grass carp in eight different tissues including the spleen [24]. There were 169 known miRNAs and 380 novel miRNAs obtained in 11 grass carp tissues including the spleen with high-throughput sequencing [25]. A total of 1208 miRNAs were identified from grass carp spleen at 0, 1, 3, 5, 7, and 9 days post-infection with a grass carp reovirus, of which 36 miRNAs exhibited differential expression [26]. In addition, 185 miRNAs associated with aeromonad septicemia in grass carp were identified from five grass carp tissues including the spleen at 4 h, 1 day, 3 days and 7 days post-infection with motile *Aeromonas hydrophila* [27]. While providing a basis for understanding the molecular biology of grass carp spleen, these studies investigated the spleen under host-pathogen conditions and were not specific to the spleen. Hence, no studies have reported miRNA transcriptomic profiling, and interaction networks between miRNAs in normal spleen tissue of grass carp.

The production of commercial culinary fish is an important aspect of grass carp aquaculture. During the production stage, grass carp grow rapidly and they are often susceptible to disease outbreaks. This is primarily due to factors such as aquaculture water quality and pathogenic microorganisms. Therefore, discovering the molecular biological characteristics of immune defense functions in the grass carp immune system will help to improve the control and prevention of disease. Considering the importance of the spleen in immune defense, and the important role of miRNAs in the regulation of spleen function, two miRNA libraries were constructed from the spleen tissue of one-year-old and three-year-old healthy grass carp. Through high throughput sequencing and subsequent bioinformatic analysis, the characterization of miRNA transcriptome profiling in normal spleen tissues of grass carp was described. Likewise, the miRNA regulatory network related to the spleen's physiological function was established. The main aim of this work is to provide new insight into the molecular biological characteristics of the spleen's physiological function. This will prove to be a valuable resource for future research on the role of miRNAs in the defense function of spleen in grass carp.

## Materials and methods

### Ethics statement

Experimental and animal care was executed in accordance with the program approved by the Experimental Animal Management Ordinance (Ministry of Science and Technology, China, 2004), and was approved by the Institutional Animal Care and Use Committee of Henan Agricultural University in China.

### Sample collection and RNA isolation

The experimental animals in this study were the cultivated grass carp *Ctenopharyngodon idella* from the Animal Center of Henan Agricultural University. A total of 120 experimental fish at one-year (cis1) and three-year of age (cis3) were taken to the laboratory and temporarily raised under standard conditions for one week. Nine healthy individuals from each developmental stage were randomly selected for spleen tissue collection. The experimental fish were euthanized by tricaine methanesulfonate (MS-222) immersion according to the recommendations on the Operational Specification for the Euthanasia of Laboratory Fishes (local standard of Shandong Province, China; DB37/T 4134–2020). Firstly, 2000 mg/L concentrated stock solutions of MS-222 (Sigma Chemical Company, St Louis, MO, USA) were prepared and stored in amber jars at 4˚C. Then, a suitable amount of concentrated stock solutions were taken and further diluted with water from the fish's environment to a final concentration of 350 mg/L.

Subsequently, anesthetic solution (350 mg/L MS-222) was placed in a tank, and fish were completely immersed in the anesthetic solution for 20 mins. Once swimming and opercular movement had completely stopped, fish were removed from the anesthetic tank and transferred to the recovery tank filled with water from the fish's environment. From euthanized fish, samples of spleen tissues were immediately collected and frozen in liquid nitrogen. Total RNA was isolated from spleen tissues using Trizol reagent (TaKaRa, Dalian, China) according to the manufacturer's instructions. The spleen tissues were homogenized by grinding in a mortar. The RNA concentrations were determined by NanoDrop 2000 spectrophotometry (Thermo Scientific, Wilmington, DE, USA), the RNA integrity was assayed by agarose gel electrophoresis. The 260/280 ratios and RIN (RNA Integrity Number) values of total RNA samples were 1.701~2.016 and 7.8~8.4, respectively, meeting the requirements of subsequent experiments.

### MiRNA library construction and post-sequencing analysis

To characterize a general overview of miRNAs expressed in grass carp spleen tissue during normal development, the high-quality RNA samples from spleen tissues of three individuals were randomly mixed equally at each developmental stage, and the mixed RNA sample was used for cDNA library construction. Two miRNA libraries were constructed from the spleen tissue of grass carp at one year and three years of age using the Truseq$^{TM}$ Small RNA sample prep Kit (Illumina, San Diego, USA). The initial total RNA was 1μg for each miRNA library construction. The cDNA library was purified by 6% Novex TBE PAGE gel (Invitrogen, CA, USA) and enriched by the bridge PCR amplification using cBot Truseq SR Cluster Kit v3-cBot-HS (Illumina, San Diego, USA) and quantified using the TBS380 Picogreen (Invitrogen, CA, USA). Finally, the qualified cDNA library was sequenced using Hiseq4000 Truseq SBS Kit v3-HS (Illumina, San Diego, USA) on the Illumina HiSeq 4000 sequencing platform.

Raw reads were obtained by transforming the Illumina sequencing image data and cleaned using SeqPrep (Releases 34, https://github.com/jstjohn/SeqPrep) and Sickle (Releases 4, https://github.com/najoshi/sickle) software by removing low quality reads, and adaptor sequences. After elimination of redundancy, the 18–32 nt clean reads, called small RNA (sRNA), were used for subsequent analyses. For these clean reads, the reads with identical sequences were merged to obtain the unique sequence (that is, unique small RNA), which were used to count the sRNA species in the samples as well as the common and unique sequences between the samples. The length distribution of clean reads was calculated using the Fastx-Toolkit software (Version 0.0.13, http://hannonlab.cshl.edu/fastx_toolkit/). The clean reads were blasted against RNA families in the Rfam database (11.0, http://Rfam.sanger.ac.uk/) to discard ribosomal RNA (rRNA), small cytoplasmic RNA (scRNA), small nucleolar RNA (snoRNA), small nuclear RNA (snRNA), and transfer RNA (tRNA). Subsequently, the remaining sRNA sequences were mapped to the reference genome using Bowtie software (Releases 1.2.3) [28] to analyze the distribution of sRNAs on the reference sequence. Due to the lack of available genomic information in grass carp, the genome of zebrafish with close relationships was used as a reference in this study.

### Identification of miRNAs

The conserved miRNAs in the spleen were identified in grass carp. All clean sequences after removing non-miRNA sequences were searched against the sequences of mature metazoan miRNAs in the miRBase (Release 22.1) [29]. The sRNA sequences with identical or related sequences (no more than one mismatch in the seed region or a few end nucleotide variations in the entire length) to the reference mature miRNAs were mapped to reference genome

using the Bowtie software (Releases 1.2.3) [28]. The 300 base sequence flanking each sRNA sequence was extracted as the potential precursor of miRNAs, and their folding secondary structures were evaluated using RNAfold (Version 2.4.13) [30] and miRDeep2 (Version 0.1.3) [31]. Based on the above analysis, the sequences that matched a miRNA in the miRBase, and whose precursor could form a typical stem-loop hairpin structure, were identified as a known grass carp conserved miRNA. In addition, all unannotated sequences were put into the miR-Deep2 modules program and used to identify potential novel miRNAs by miRDeep2 under the system default parameter. These novel miRNAs were named as described by Ambros et al. (2003) [32].

## Identification of significantly differentially expressed miRNAs

The expression levels of the identified miRNAs were estimated using transcripts per million clean reads (TPM). Based on the normalized TPM values, the differentially expressed miRNAs were analyzed by DEGseq (Releases 3.12) [33]. The fold-change for each miRNA between one-year and three years of age was calculated. The $p$-value was determined by Fisher's exact test and adjusted using $q$-value [34]. The $q$-value $<0.05$ and |log2 (fold change)| $>1$ were set as the threshold for SDE miRNAs.

## MiRNA target prediction, functional enrichment analysis and interaction analysis between miRNAs and mRNAs

To predict the potential target genes of miRNAs, transcriptome profiles of spleen tissue from grass carp one year and three years of age were also constructed using the same tissue samples and the miRNA libraries [35]. The raw sequencing data had been deposited into the NCBI database Sequence Read Archive under number SRP078553. Based on the above mentioned transcriptome profile data, the potential target genes of the SDE miRNAs were predicted using the miRanda (Releases 3.3a) [36] and TargetScan (Releases 6.2) [37] software. When the TargetScan software was used to predict the target genes, only the sites belonging to the top quartile of ranked predictions, present in zebrafish, mouse, and human, were selected as the true binding sites. For the miRanda software, all detected targets with the threshold parameters of S $>140$ and $\Delta G <$ -10 kcal/mol and strict 5' seed pairing were considered as potential targets. In order to make the identification of target genes robust, the target genes that were simultaneously predicted by two programs were selected for subsequent functional enrichment analysis. GO and KEGG pathway enrichment analyses were performed using the Goatools (Version v0.7.6) [38] and KOBAS (Version 3.0) [39] software, respectively. The corrected $p$-value $\leq 0.05$ was used as the threshold for significant enrichment of a GO term or pathway. Based on the results of the KEGG pathway enrichment analysis, the negatively correlated SDE miRNA-mRNA pairs expressed in the spleen of grass carp were selected from the pathways associated with immunity, lipid metabolism, and cell proliferation and differentiation, and their interaction networks were constructed using Cytoscape software (Version 3.2) (http://www.cytoscape.org/).

## Quantitative real-time PCR (qRT-PCR) analysis

Using the qRT-PCR analysis, six SDE miRNAs including miR-18c, miR-451, miR-144-5p, miR-100-3p, miR-363-3p and miR-2188-5p were validated in spleen from one year and three years of grass carp. The qRT-PCR was performed using the customized kits containing the stem-loop reverse transcription primer and miR-specific molecular beacon probe (RiboBio, Guangzhou, China) on the LightCycler® 96 instrument (Roche, Basel, Switzerland). The total RNA from spleen tissues were treated with DNase I (RNase-free) (TaKaRa, Dalian, China),

and then reverse-transcribed with M-MLV Reverse Transcription Reagents (Invitrogen, CA, USA) and the stem-loop reverse transcription primer according to the manufacturer's instructions. The PCR amplification reaction system was as follows: 1 μL cDNA product, 5 μL 2×SYBR Premix Ex Taq II, 1 μL specific primer (10 μmol/L), and 3 μL ddH$_2$O. The PCR amplification program for these miRNAs was as follows: 95˚C for 3 mins; 40 cycles of 95˚C for 12 s, 60˚C for 40 s, and 72˚C for 30 s; and 10 mins extension at 72˚C. All qRT-PCR analyses included three biological replicates, and each biological replicate contained three technical replicates. U6 small nuclear RNA was used as endogenous control, and the relative expression levels were calculated by the $2^{-\Delta\Delta ct}$ method [40]. The t-test was used to determine the $p$-value, and $p$-value$<0.05$ was considered to be a significant difference.

## Statistical analysis

The qRT-PCR data and graphs were generated in GraphPad Prism (version 5.0) software (San Diego, CA, USA). Statistical significance of the qRT-PCR results was tested by performing two-tailed unpaired t-tests. Data are presented as the means containing three replicates.

## Results

### MiRNA library sequencing and sequence analysis

Through Illumina HiSeq sequencing, 15,839,580 and 16,587,512 high-quality clean reads with a length range of 18–32 nt were obtained from spleen tissue of grass carp at one year (cis1) and three years (cis3) of age, respectively. The length of these cleaned reads consisted of 20–23 nucleotides, and peaked at 22 nt. This represented approximately 35% of the total number of clean reads in the libraries (**S1A Fig**). The statistical results showed that a total of 31,811,397 sRNAs representing 100,557 unique sRNAs were shared in the cis1 and cis3 libraries (**S1B and S1C Fig**). In addition, 293,603 sRNAs were annotated with the RFam 11.0 database, accounting for 86.49% of the total number of clean reads. In particular, the number of matched miRNA sequences reached 80% (**S1D Fig**), indicating a high miRNA abundance in the library. However, the matched unique sRNAs in the cis1 and cis3 libraries only accounted for 23.01% and 19.14%, respectively, indicating that there were many specific sRNAs in the spleen of grass carp. All clean sRNA sequences, after removing rRNA, tRNA, snoRNA, and other snRNAs, were aligned with the reference genome sequence, and a total of 26,731,633 (87.92%) reads were perfectly mapped. Among them, the number of mapped sequences on chromosome 14 (25.02%) was the largest, followed by chromosome 8 (13.37%), chromosome 2 (12.14%), chromosome 23 (11.25%), and chromosome 15 (10.98%) (**S1E Fig**).

### MiRNAs expressed in grass carp spleen tissue

To identify conserved miRNAs in the spleen of grass carp, all clean sequences mapped to reference genomes were compared to the mature sequences of miRNAs from all animals in the miRBase (release 22.1). A total of 26,022,975 (80.25%) reads with homology to known miRNAs were annotated as conserved miRNAs. When combined with the structural predictions of precursor sequences, 324 known conserved miRNAs were identified in grass carp spleen tissue from two developmental stages (**S1 Table**). Of the 324 conserved miRNAs, 132 miRNAs were perfectly matched with known miRNAs of zebrafish. However, 157 miRNAs were matched with known zebrafish miRNAs but showed variation at the end of mature sequences or differences in genomic localization of precursor sequences. The remaining 35 miRNAs were homologous with known miRNAs from 24 animals, including *Cyprinus carpio*, *Ictalurus punctatus*, *Oryzias latipes*, *Salmo salar*, *Ciona intestinalis*, *Ornithorhynchus anatinus*, *Taeniopygia gu*ttata,

*Ascaris suum*, *Bombyx mori*, *Caenorhabditis elegans*, *Cricetulus griseus*, *Drosophila melanogaster*, *Drosophila willistoni*, *Eptesicus fuscus*, *Equus caballus*, *Homo sapiens*, *Macaca mulatta*, *Monodelphis domestica*, *Mus musculus*, *Pan troglodytes*, *Pristionchus pacificus*, *Rattus norvegicus*, *Schistosoma mansoni*, and *Tupaia chinensis*. In addition, sRNAs sequences that did not match known miRNAs were compared with the reference genome sequence, and novel miRNAs were predicted by miRDeep2 according to the criteria of potential miRNAs identity mentioned above. Ultimately, nine novel miRNAs were identified in the spleen of grass carp (**S2 Table**).

In addition, a family analysis of 324 known miRNAs was carried out using the sequence similarity searches. The results showed that 300 miRNAs belonged to 105 miRNA gene families. Among these miRNA families, some families such as let-7, mir-1, mir-10, mir-124, mir-8, mir-7, mir-9, mir-153, mir-190, mir-137, and mir-25 were present both in invertebrates and vertebrates, while fourteen miRNA families existed only in fish. These were possibly fish-specific miRNAs that included mir-459, mir-430, mir-462, mir-7147, mir-2187, mir-722, mir-724, mir-737, mir-725, mir-728, mir-729, mir-731, mir-734, and mir-738 (**Fig 1**). More than two family members in 64 miRNA families were expressed in the spleen of grass carp. In particular, the miRNA families mir-10, let-7, mir-15, mir-17, mir-130, mir-30 and mir-8 had twenty, thirteen, twelve, eleven, nine, nine, and nine members, respectively.

## The expression characteristic of miRNAs in grass carp spleen tissue

Among the 324 known miRNAs identified in this study, 308 miRNAs were shared in two development stages of the spleen, seven miRNAs were expressed only in the spleen of one-year-old grass carp, and nine miRNAs were expressed specifically in the spleen of three-year-old grass carp. The distribution of miRNA expression levels showed that the expression pattern of miRNAs was highly consistent between one-year-old and three-year-old spleen (**S2A–S2C Fig**). In addition, the abundance of these identified miRNAs was different in the grass carp spleen. The percentages of miRNAs with TPM > 100 in the cis1 and cis3 were 40.7% and 39.2%, respectively. These abundant miRNAs were primarily concentrated in families such as let-7, mir-10, mir-30, and mir-15, which are known to be involved in immunity (**Table 1**).

The expression levels of spleen miRNAs in grass carp at one year and three years of age were compared (**S2D Fig**). Under the criteria of |log2(foldchange)|>1 and *q*-value <0.05, 20 SDE miRNAs were identified compared with one-year old spleen (**S2E Fig** **and S3 Table**). Surprisingly, only cid-miR-212-5p and cid-miR-100-3p among these SDE miRNAs were upregulated. However, most miRNAs with high abundance were not differentially expressed between the two development stages in the spleen. In addition, six SDE miRNAs including cid-miR-144-5p, cid-miR-100-3p, cid-miR-451, cid-miR-18c, cid-miR-2188-5p, and cid-miR-363-3p were verified by qRT-PCR analysis. Compared with the RNA-seq data, the expression levels determined by qRT-PCR showed similar expression patterns (**Fig 2**), indicating that the sequencing analysis results were reliable in this study.

## The potential function of SDE miRNAs

Based on the spleen transcriptome data, a total of 6,887 potential target genes of the SDE miRNAs were predicted in grass carp spleen tissue. Subsequently, the KEGG pathway enrichment analyses for the predictive target genes were performed, and 37 significantly enriched pathways were identified. These pathways were clustered into seven classes including environmental information processing, genetic information processing, cellular processes, organismal systems, drug development, human diseases and metabolism (**Fig 3**). Additionally, some of these pathways were involved in immunity and lipid metabolism, such as the B cell receptor

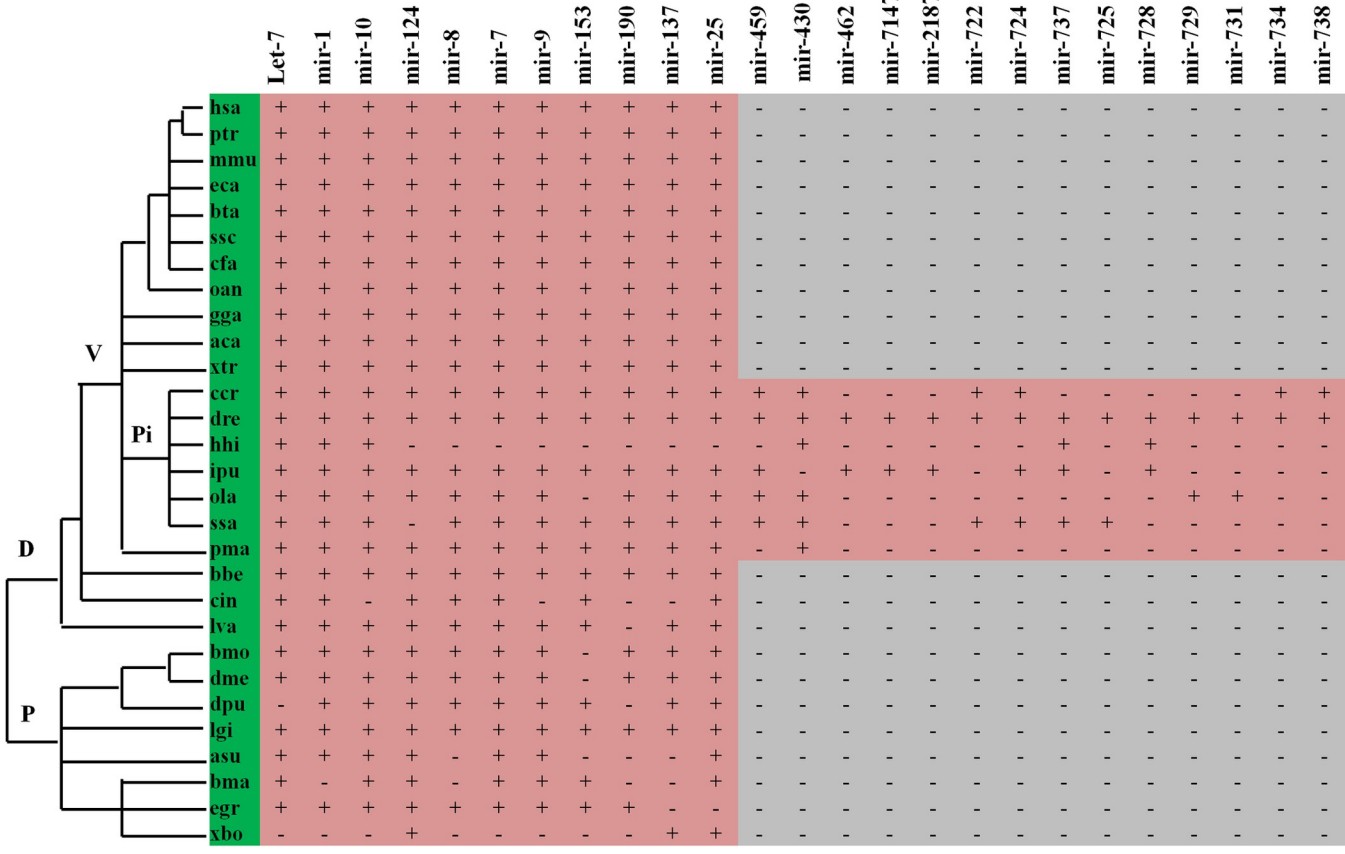

**Fig 1. Highly conserved and fish-specific miRNA family in grass carp spleen.** The presence of miRNA is indicated by a plus (+); the absence of miRNA is indicated by a minus (-). Abbreviations: hsa, *Homo sapiens*; ptr, *Pan troglodytes*; mmu, *Mus musculus*; eca, *Equus caballus*; bta, *Bos taurus*; ssc, *Sus scrofa*; cfa, *Canis familiaris*; oan, *Ornithorhynchus anatinus*; gga, *Gallus gallus*; aca, *Anolis carolinensis*; xtr, *Xenopus tropicalis*; ccr, *Cyprinus carpio*; dre, *Danio rerio*; hhi, *Hippoglossus hippoglossus*; ipu, *Ictalurus punctatus*; ola, *Oryzias latipes*; ssa, *Salmo salar*; pma, *Petromyzon marinus*; bbe, *Branchiostoma belcheri*; cin, *Ciona intestinalis*; lva, *Lytechinus variegatus*; bmo, *Bombyx mori*; dme, *Drosophila melanogaster*; dpu, *Daphnia pulex*; lgi, *Lottia gigantea*; asu, *Ascaris suum*; bma, *Brugia malayi*; egr, *Echinococcus granulosus*; xbo, *Xenoturbella bocki*. P, protostomia; D, deutostomia; V, vertebrata; Pi, pisces.

signaling pathway, T cell receptor signaling pathway, insulin signaling pathway, PPAR signaling pathway, and fatty acid biosynthesis. Moreover, GO enrichment analysis of these target genes showed that 1,319, 301 and 790 GO terms were significantly enriched in biological process, cellular component and molecular function, respectively. Functions of the SDE miRNAs were concentrated in the ubiquitin-dependent protein catabolic process, phospholipid metabolic process, macromolecule catabolic process, chromatin modification, positive regulation of cellular metabolic process, endoplasmic reticulum membrane, bounding membrane of organelle, ubiquitinyl hydrolase activity, and guanyl-nucleotide exchange factor activity (**Fig 4A**). In particular, many significant enrichment GO terms were related to hematopoietic, immune response and lipid metabolism in the category of biological process, for instance, erythrocyte differentiation, hematopoietic or lymphoid organ development, immune response, immune system processes, lipid metabolic processes, and fatty acid metabolic processes (**Fig 4B**).

## Interaction networks of mRNAs in grass carp spleen

To further reveal the potential regulatory mechanisms of these SDE miRNAs, negatively correlated SDE miRNA-mRNA pairs were identified based on the KEGG enrichment analysis results for the SDE miRNA prediction target genes, and then miRNA-mRNA interaction

**Table 1. The abundance and function of miRNAs with TPM > 1000 in grass carp spleen tissue.**

| MiRNA family | MiRNA name | TPM value in cis1 | TPM value in cis3 | The function of miRNA family member and reference |
|---|---|---|---|---|
| let-7 | cid-let-7a | 70576.41 | 64017.30 | Innate and adaptive immune responses [41] |
| | cid-let-7e | 17646.64 | 19763.12 | |
| | cid-let-7g | 11432.37 | 11702.26 | |
| | cid-let-7f | 6810.24 | 7809.72 | |
| | cid-let-7h | 2081.93 | 2201.85 | |
| | cid-let-7i | 1580.80 | 1016.29 | |
| | cid-let-7j | 1578.86 | 1750.40 | |
| | cid-let-7d-5p | 1318.24 | 1124.60 | |
| | cid-let-7b | 1104.70 | 983.65 | |
| | cid-let-7c-5p | 1027.07 | 1044.89 | |
| mir-10 | cid-miR-10c-5p | 54711.66 | 56931 | Innate immunity and inflammation [42]; T-cell activation and cytotoxicity [43] |
| | cid-miR-100-5p | 9619.79 | 9529.29 | |
| | cid-miR-125a | 2092.06 | 1933.15 | |
| | cid-miR-10b-5p | 1678.33 | 1882.18 | |
| | cid-miR-125b-5p | 1563.21 | 1623.13 | |
| | cid-miR-99 | 1277.11 | 1355.12 | |
| | cid-miR-10a-5p | 831.86 | 1071.69 | |
| mir-30 | cid-miR-30d | 16688.71 | 16454.72 | Innate immunity during macrophage and dendritic cell differentiation [44] |
| | cid-miR-30e-5p | 12525.31 | 11275.32 | |
| | cid-miR-30c-5p | 3962.83 | 3590.57 | |
| | cid-miR-30b | 2550.87 | 2151.79 | |
| | cid-miR-30e-3p | 1236.06 | 1155.29 | |
| mir-126 | cid-miR-126a-5p | 14291.10 | 10858.03 | Activation and function of CD4+ T cells [45]; Modulator of innate immunity [46] |
| | cid-miR-126b-5p | 14291.10 | 10858.03 | |
| | cid-miR-126a-3p | 6660.78 | 5821.52 | |
| | cid-miR-126b-3p | 1093.68 | 889.36 | |
| mir-15 | cid-miR-16b | 4054.10 | 3635.36 | The switch from pre-B cell proliferation to differentiation [47]; Memory T Cell Differentiation [48] |
| | cid-miR-15b-5p | 3489.12 | 3298.82 | |
| | cid-miR-16c-5p | 2599.30 | 2340.09 | |
| | cid-miR-457a | 1135.10 | 1021.91 | |
| mir-221 | cid-miR-221-3p | 1419.57 | 1286.80 | Innate anti-viral response [49] |
| | cid-miR-222b | 1347.97 | 1453.23 | |
| | cid-miR-222a-3p | 1062.09 | 725.71 | |
| | cid-miR-222a-5p | 979.91 | 1533.28 | |

(*Continued*)

**Table 1.** (Continued)

| MiRNA family | MiRNA name | TPM value in cis1 | TPM value in cis3 | The function of miRNA family member and reference |
|---|---|---|---|---|
| mir-142 | cid-miR-142a-5p | 32490.65 | 23792.57 | Innate immunity during macrophage and dendritic cell differentiation [44]; Phagocytosis of myeloid inflammatory cells [50]; CD25$^+$ CD4 T cell proliferation [51] |
| | cid-miR-142b-5p | 5974.80 | 5628.28 | |
| | cid-miR-142a-3p | 2179.83 | 1735.05 | |
| mir-181 | cid-miR-181a-5p | 44088.28 | 49097.67 | T cell activation and actin polymerization-mediated T cell functions [52]; Early iNKT cell development [53] |
| | cid-miR-181a-3p | 2860.96 | 5182.18 | |
| | cid-miR-181b-5p | 1460.17 | 1425.31 | |
| mir-199 | cid-miR-199-3-3p | 1506.14 | 1538.14 | Neutrophil migration [54] |
| | cid-miR-199-3p | 1405.26 | 1324.36 | |
| | cid-miR-199-5p | 1354.89 | 1237.71 | |
| mir-144 | cid-miR-144-5p | 4724.66 | 1730.75 | Inflammatory factor secretion from macrophages [55] |
| | cid-miR-144-3p | 3317.98 | 1091.20 | |
| mir-145 | cid-miR-145-5p | 1673.56 | 1843.02 | Macrophage polarization [56] |
| | cid-miR-145-3p | 1464.57 | 1427.88 | |
| mir-146 | cid-miR-146a | 8566.79 | 7039.78 | Innate immunity and inflammation [42]; Innate immune sensing [57]; Hematopoiesis and immune function [58] |
| | cid-miR-146b | 3405.53 | 3258.06 | |
| mir-25 | cid-miR-25-3p | 63066.19 | 48310.44 | Hypoxia-induced immunosuppression [59] |
| | cid-miR-92a-3p | 29581.48 | 20523.47 | |
| mir-27 | cid-miR-27b-3p | 7215.70 | 7981.71 | Treg-mediated immunological tolerance [60]; Effector T cell differentiation and function [61] |
| | cid-miR-27c-3p | 2810.60 | 2722.66 | |
| mir-101 | cid-miR-101a | 12678.56 | 9155.74 | Innate immune responses of macrophages [62] |
| mir-128 | cid-miR-128-3p | 1036.08 | 1060.38 | Inhibit common lymphoid progenitors from developing into progenitor B cells [63] |
| mir-143 | cid-miR-143 | 222118.58 | 288323.88 | Memory T cell differentiation [64] |
| mir-148 | cid-miR-148 | 1078.03 | 930.61 | Inflammatory response to bacterial infection [65]; Innate response and antigen presentation of TLR-triggered dendritic cells [66] |
| mir-150 | cid-miR-150 | 2141.01 | 1068.92 | Memory CD8 T cell differentiation [67]; Negatively regulate CD4(+) T cell function [68]; Development of NK and iNKT cells [69] |
| mir-192 | cid-miR-192 | 6197.20 | 1933.35 | Immune response after pathogen infection [70] |
| mir-21 | cid-miR-21 | 86621.38 | 93020.24 | Suppress cytokines production [71]; T lymphocyte activation [72] and apoptosis [73] |
| mir-210 | cid-miR-210-3p | 2285.33 | 2440.98 | Antiviral innate immune response [74]; Respiratory burst [75]; B cells and autoantibody production [76] |
| mir-223 | cid-miR-223 | 3313.21 | 4900.77 | Regulator of innate immunity [77]; Type I interferon production in antiviral innate immunity [78] |
| mir-29 | cid-miR-29a | 1040.11 | 994.14 | Interferon-γ production in helper T cells [79]; Innate and adaptive immune responses to intracellular bacterial infection [80] |
| mir-451 | cid-miR-451 | 36096.07 | 16792.92 | CD4$^+$T cell proliferative responses to infection [81] |

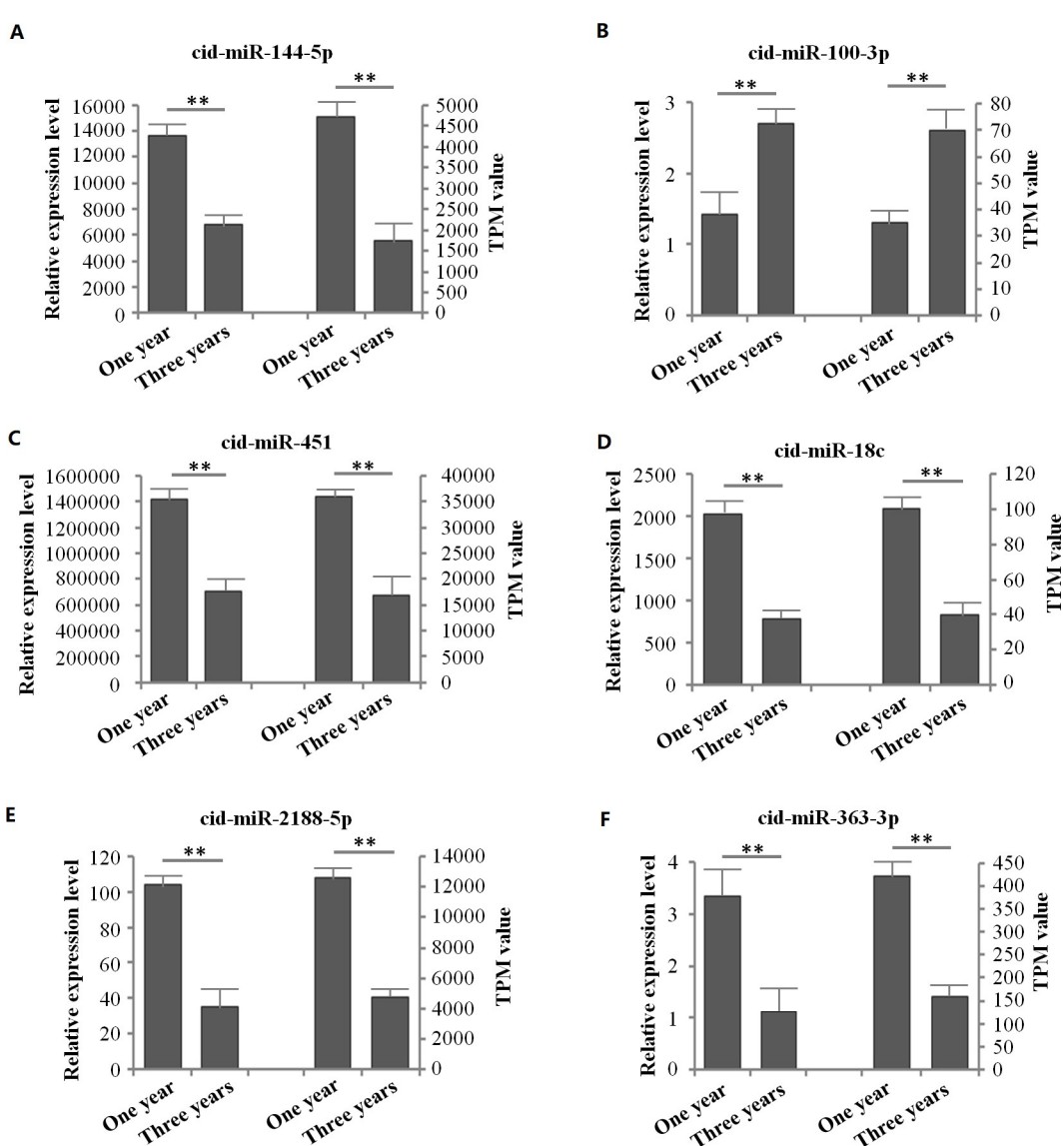

**Fig 2. The qRT-PCR verification of the selected SDE miRNAs in the spleen of grass carp at one year and three years of age.** The A, B, C, D, E and F in figure are the qRT-PCR results and TPM value of RNA-seq corresponding to cid-miR-144-5p, cid-miR-100-3p, cid-miR-451, cid-miR-18c, cid-miR-2188-5p and cid-miR-363-3p, respectively. Data are represented by the mean ± SEM (n = 9) in qRT-PCR results. $^{**}$p-value < 0.01 represent significant difference.

networks were constructed by Cytoscape software (Version 3.2). Thereinto, the immune-related interaction network contains 10 miRNA-mRNA pairs (Fig 5), which were screened from 13 immune and hematopoietic pathways including toll-like receptor signaling pathway, RIG-I-like receptor signaling pathway, NOD-like receptor signaling pathway, cytosolic DNA-sensing pathway, T cell receptor signaling pathway, B cell receptor signaling pathway, hemato-poietic cell lineage, Fc epsilon RI signaling pathway, complement and coagulation cascades, natural killer cell mediated cytotoxicity, leukocyte transendothelial migration, chemokine signaling pathway, and platelet activation. In this network, cid-miR-144-5p is the key node of this network. The gene *NFKBA* targeted by cid-miR-144-5p is a common component of seven pathways. Similarly, another gene *P38* targeted by cid-miR-144-5p is also a common

## The Most Enriched GO Terms

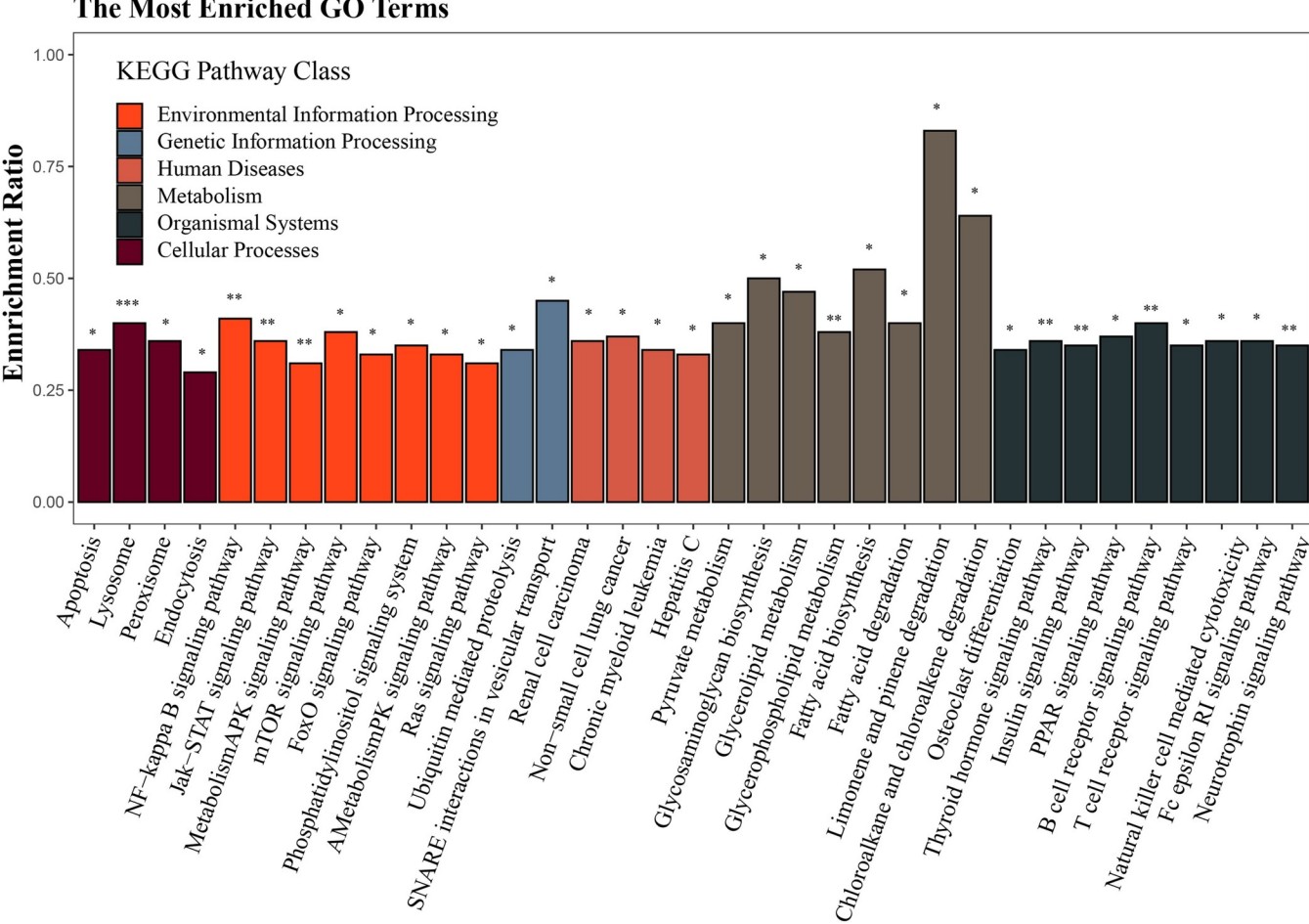

**Fig 3. The significantly enriched pathways for potential genes targeted by SDE miRNAs in grass carp spleen.** The abscissa indicates the enriched KEGG pathways and class. The ordinate indicates the enrichment ratio in each KEGG pathway, and the enrichment ratio is calculated as follows: sample number/ background number. ***$p$-value < 0.001; **$p$-value < 0.01; *$p$-value < 0.05.

component of seven pathways. Therefore, cid-miR-144-5p appears to have a wide range of regulatory effects. In contrast, other miRNAs in this network have specific roles.

The development of spleen is accompanied by a series of cell proliferation and differentiation events. To this end, a miRNA-mRNA interaction network associated with cell proliferation and differentiation was constructed (**Fig 6**), which contained 28 miRNA-mRNA pairs involving TNF signaling pathway, MAPK signaling pathway, PI3K-Akt signaling pathway, AMPK signaling pathway, Ras signaling pathway, Jak-STAT signaling pathway, NF-kappa B signaling pathway, mTOR signaling pathway, ECM-receptor interaction, cGMP-PKG signaling pathway, cAMP signaling pathway, FoxO signaling pathway, VEGF signaling pathway, and Rap1 signaling pathway. In this network, genes such as *SOCS3*, *LAMA3_5*, *DDIT4*, *ATP1A*, *TNFB*, *HGF*, *NFKBIA*, *CFLAR*, *P38*, and *RASGRF2* are involved in more than two pathways. Likewise, cid-miR-122, cid-miR-18c, cid-miR-20b-5p, cid-miR-205-5p, and cid-miR-363-3p, as well as members of mir-144, mir-203, and mir-9 families, all acted on more than two genes. These miRNAs, together with their predicted target genes, constituted the key nodes of this network and appear to have a broad range of functions. In contrast, some miRNAs, such as cid-miR-182-5p and cid-miR-375 in the MAPK signaling pathway, cid-miR-725-

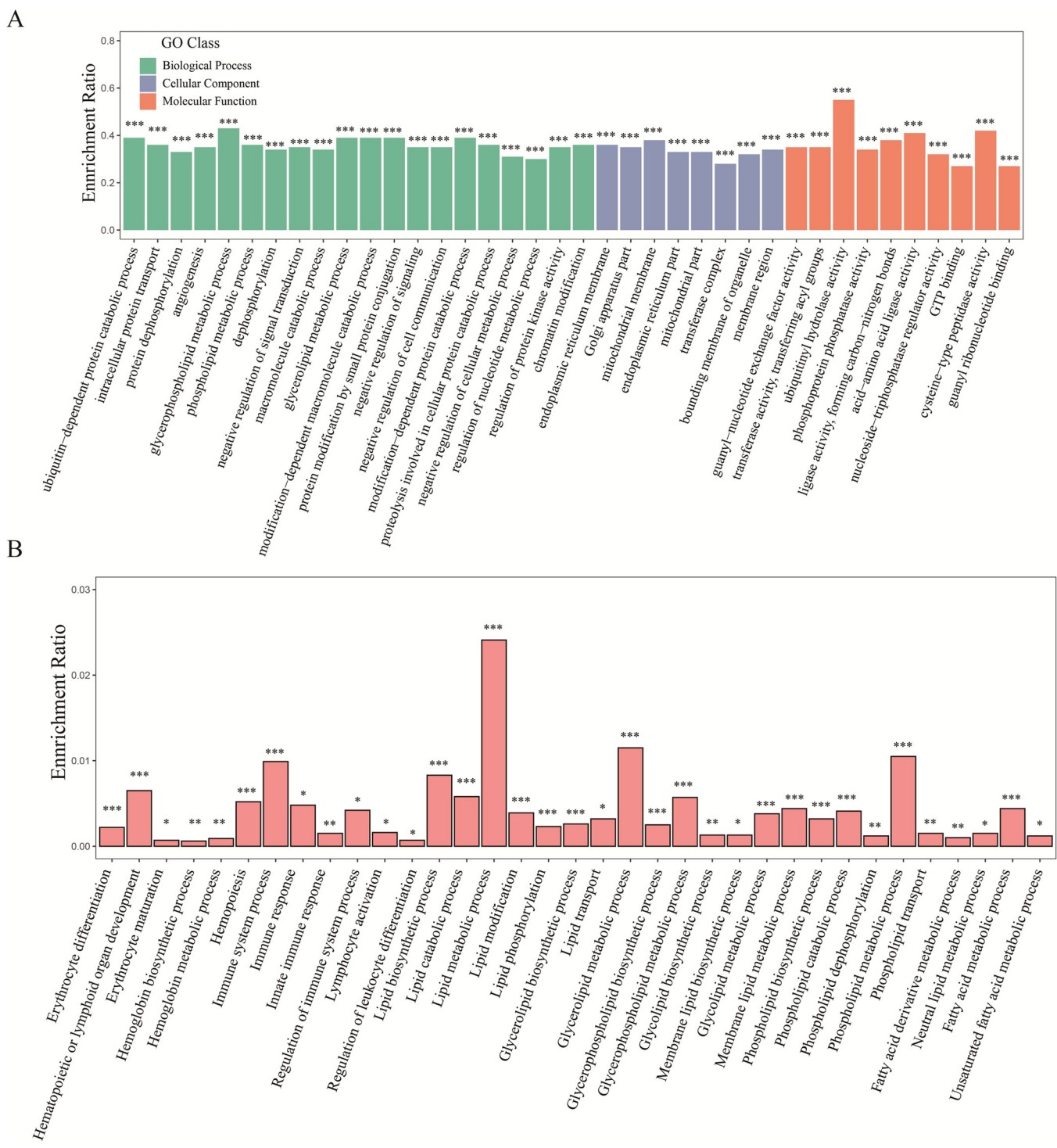

**Fig 4. The significantly enriched GO terms for potential genes targeted by SDE miRNAs in grass carp spleen.** (A) Part of significantly enriched GO terms in the category of biological process, cellular component, and molecular function. (B) Significantly enriched GO terms associated with immunity and lipid metabolism in the category of biological process. The abscissa indicates the enriched GO terms and category. The ordinate indicates the enrichment ratio for each GO term, and the enrichment ratio is calculated as follows: sample number/background number. ***$p$-value < 0.001; **$p$-value < 0.01; *$p$-value < 0.05.

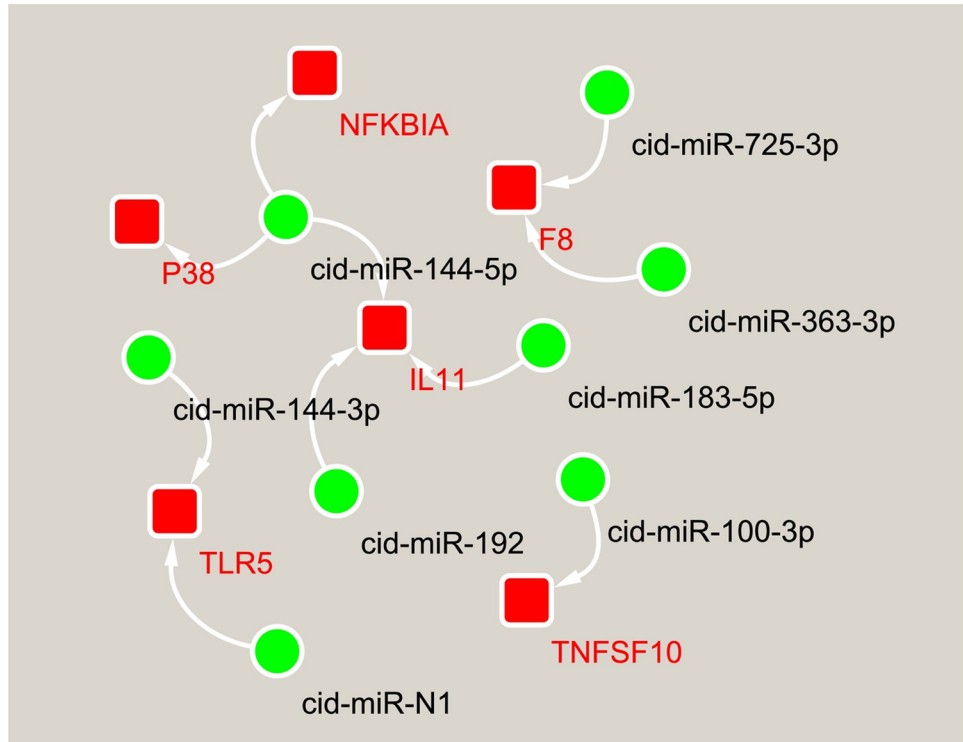

**Fig 5. The miRNA-mRNA interaction networks are associated with immunity and hematopoiesis.** The dot indicates miRNA, and the box indicates a target gene that had a negative correlation with a given miRNA.

3p in the AMPK signaling pathway, and cid-miR-216a in the FoxO signaling pathway, which target only one gene in this network and have specific roles.

In addition, we found that the predicted target genes of SDE miRNAs were significantly enriched in some pathways associated with lipid metabolism. Hence, an interaction network was constructed using 23 miRNA-mRNA pairs selected from five pathways, such as fatty acid metabolism, glycerolipid metabolism, insulin signaling pathway, adipocytokine signaling pathway, and PPAR signaling pathway (Fig 7). In this network, cid-miR-182-5p, cid-miR-18c, cid-miR-725-3p, cid-miR-144-5p, and members of the mir-203 family all target more than two genes. Similarly, *SOCS3*, *SCD*, and *glpK* are common components of two pathways and were targeted by three miRNAs. These genes became the interrelated centers in the miRNA-mRNA network maintaining collectively the stability of this interaction network. It is noteworthy that the gene *PPP1R3* in the insulin signaling pathway was targeted by four miRNAs, and the gene *ACSL* was a common component of fatty acid metabolism, the PPAR signaling pathway, and the adipocytokine signaling pathway targeted by three miRNAs.

## Discussion

Due to the pivotal role played in modulating immune responses, the spleen is gaining more attention. In the past decade, spleen research has focused on morphology, organogenesis, and function as well as the genetic regulation underlying these processes. In recent years, with the rapid development and application of high-throughput sequencing technologies, considerably more attention has been focused on elucidating the molecular mechanisms regulating spleen function. The primary goal is to identify genes involved in spleen function by transcriptome

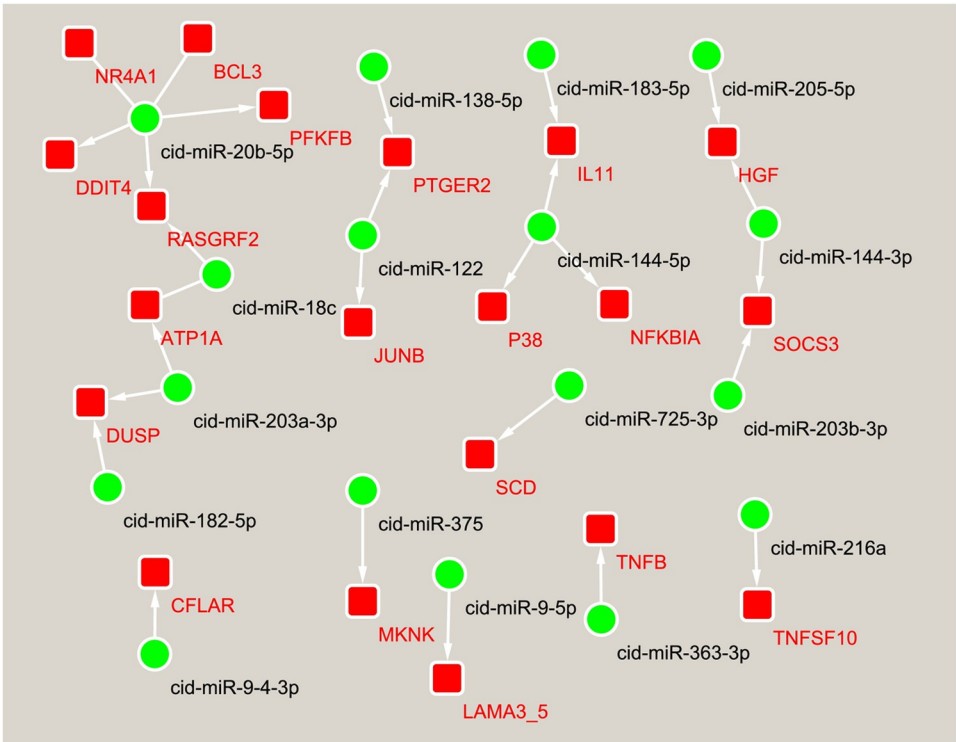

**Fig 6. The miRNA-mRNA interaction networks are associated with cell proliferation and differentiation.** The dot indicates miRNA, and the box indicates a target gene that had a negative correlation with a given miRNA.

analysis in different species, such as the mouse [8, 82], goose [83], pig [84], giant panda [85], common carp [86] as well as bighead carp and silver carp [87]. In addition, to elucidate epigenetic regulation of spleen function, non-coding RNAs such as miRNA, lncRNA or circRNA were identified by transcriptome sequencing in different species, such as the mouse [82], pig [83], giant panda [84], Chinese giant salamander [88], green-spotted puffer fish [89], rainbow trout [90], and common carp [19]. These studies improve our understanding of the regulation of spleen function in animals. However, some investigations are limited to spleen function under stress conditions such as pathogen infection. As such, little is known about the molecular biological mechanisms regulating the spleen's physiological function. Although the transcriptome profiles involved in grass carp spleen have been studied [35], miRNA expression profiles in normal spleen tissues have not been reported. In this study, 324 conserved miRNAs and nine novel miRNAs were identified from the spleen tissues of one-year-old and three-year-old grass carp (S2 Table). The expression profile characteristics of these miRNAs were described, and the miRNA regulatory network related to the spleen's physiological function was constructed. These results expand the number of known miRNAs associated with spleen function and provide a meaningful framework to understand the molecular mechanisms regulating spleen function in grass carp and closely related species.

Evolutionary conservation is one key characteristic of miRNAs. Many homologous miRNAs are highly conserved in sequence across metazoans [91]. In addition, some miRNA families widely exist in multiple species and exhibit either expansion or contraction in different species, thus playing the corresponding regulatory role by generating different members of miRNAs [92]. Accumulating evidence indicates that conserved miRNAs across species may have similar functions in regulating different biological processes from lower to higher

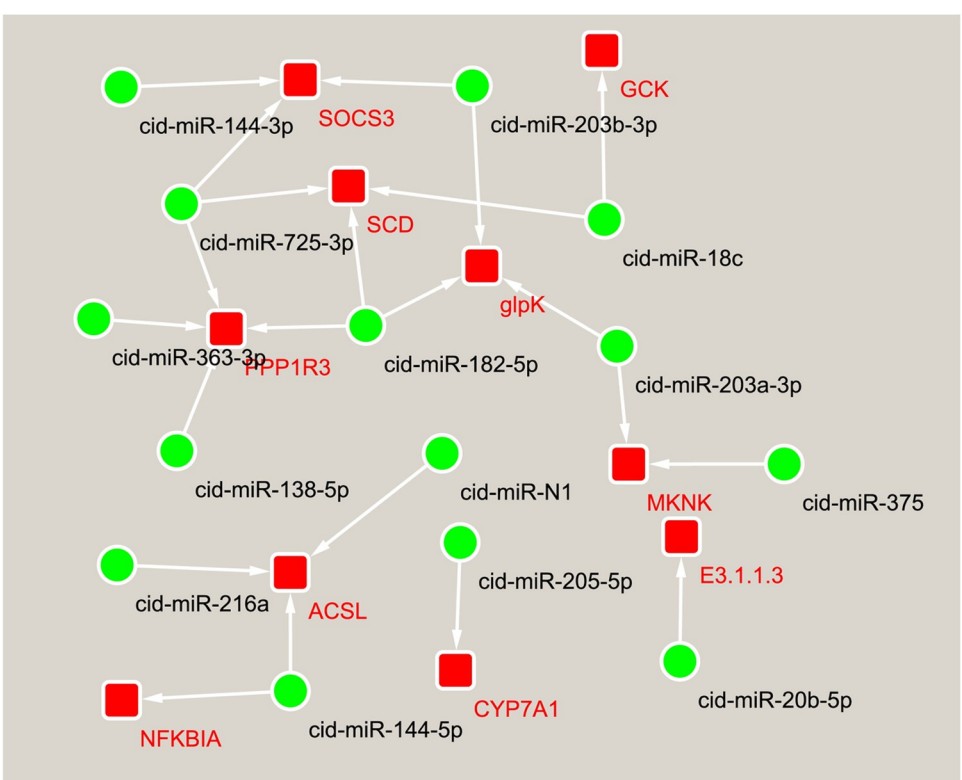

**Fig 7. The miRNA-mRNA interaction networks are associated with lipid metabolism or deposition.** The dot indicates miRNA, and the box indicates a target gene that had a negative correlation with a given miRNA.

organisms. Thus, the identification of conserved miRNAs or the conservation analysis of miR-NAs may help us infer the functions of miRNAs in a specific biological context on their known functions in other species. Due to the lack of adequate and available genomic information and the absence of known grass carp miRNAs in the miRBase database at present, the main approach for miRNA identification of grass carp was to match the sequences obtained by high-throughput screening with the known miRNA sequences of all animals in the miRBase database, thereby identifying known conserved miRNAs. Likewise, this approach was also used in this study to identify 324 known conserved miRNAs from the grass carp spleen. These identified miRNAs belong to 105 miRNA gene families, which are found in 94 species ranging from lower Platyhelminthes to the highest mammals. Among them, 28 families such as mir-1, mir-124, mir-7, and mir-9, were shared by both invertebrates and vertebrates; 63 miRNA families such as mir-103, mir-128, mir-130, mir-132, and mir-138, were present only in vertebrates; 14 families such as mir-459, mir-430, mir-462, and mir-7147, were identified only in fish (**Fig 1**). Members of 15 miRNA families, including mir-10, let-7, mir-15, mir-17, mir-130, mir-30, mir-8, mir-19, mir-153, mir-181, mir-27, mir-132, mir-25, mir-221, and mir-29, were expressed in grass carp spleen, accounting for 41% of the conserved miRNAs identified in this study. These data demonstrate that miRNAs expressed in grass carp spleen were dominated by conserved miRNAs. This is consistent with previous results that reported highly conserved miRNAs in the spleen of Chinese Giant Salamander [**88**]. Previously, the identification of miR-NAs in the spleens of Chinese giant salamander [**88**], common carp [**19**], rainbow trout [**90**] and green-spotted puffer fish [**89**] has been reported. We compared miRNAs expressed in the spleen of grass carp with those of the four species and found that 26.5%, 33.3%, 36.1% and

53.7% of the grass carp spleen miRNAs were identical to miRNAs in the spleen of Chinese giant salamander, common carp, rainbow trout, and green-spotted puffer fish, respectively, although the number of miRNAs identified by different studies varied greatly. In particular, 22 miRNAs from the families of mir-140, mir-144, mir-145, mir-181, mir-210, mir-2188, mir-221, and mir-455 were expressed in the spleen of the above five species. These studies demonstrated that miRNAs expressed in the vertebrate spleen are phylogenetically conserved. These conserved miRNAs may regulate the spleen physiology in the vertebrate. Therefore, the known conserved miRNAs identified in this study can provide a valuable resource for further research on the molecular mechanism of the spleen's physiological function in grass carp and other vertebrates.

It is estimated that miRNA genes represent approximately 1% of the known eukaryotic genomes [92]. Thus, the identification of novel miRNAs has become an essential step for understanding the variety of functions of miRNAs. Currently, next-generation sequencing such as Illumina deep sequencing is an efficient method for the identification of novel miRNAs in animals lacking adequate and available genome information. Unfortunately, only nine novel miRNAs in the grass carp spleen were identified using Illumina deep sequencing in this study. This is similar to previous studies where twenty-four and twelve novel miRNAs were obtained in the spleen of the Chinese giant salamander [88] and the common carp [19], respectively. In contrast, another study identified 926 novel miRNAs in the spleen of rainbow trout [90]. Although these studies all adopted next-generation sequencing, the number of novel miRNAs obtained was significantly different due to the use of different criteria for miRNA identification. In fact, 298 potential novel miRNAs were initially identified using the miRDeep2 software (Version 0.1.3). However, after strict sequence alignment, many of these potential novel miRNAs were found to be the length and/or end-sequence variations of known miRNA sequences identified in this study. Therefore, these potential novel miRNAs that match known miRNA but have a difference of 1–4 nucleotides at the 3' end were treated as corresponding homologue miRNAs. Finally, 157 conserved miRNAs were identified that match known zebrafish miRNAs in the miRBase database but are different at the end of mature sequences (S1 Table). The above criteria were used to identify miRNA based on the following facts: (i) the length and sequence heterogeneity of miRNA are often detected in high-throughput sequencing [19, 93–95]; (ii) the length and sequence heterogeneity in miRNA datasets are often the result of sequencing errors and post-transcriptional modifications of RNA [96]; and (iii) a difference at the last base of the miRNA between two species caused by sequencing errors is generally not treated as a change [97]. Therefore, based on the above results and analysis, it was inferred that the number of novel miRNAs or organ-specific miRNAs in the spleen of grass carp may be relatively small, and are mainly widespread and conserved miRNAs. The majority of the conserved miRNAs showed length and/or end-sequence variations that affect the formation of the miRNA/ target mRNA hybrid duplex, which increases the diversity of miRNAs and their targets [98] and thus perform distinct roles in the grass carp spleen.

The expression profile of miRNAs reflects their different roles in specific tissues or developmental stages as well as corresponding biological mechanisms. In this study, we analyzed the expression profile characteristics of miRNAs in the spleen of grass carp at two developmental stages and found that 308 known miRNAs were shared at the two developmental stages. The TPM density distribution showed that the expression levels of these miRNAs were highly consistent between one-year-old and three-year-old spleens (S2 **Fig**). This temporal expression feature of miRNAs in the grass carp spleen is completely consistent with previous results in the spleen of giant pandas [88]. In grass carp spleen, known miRNAs are highly expressed, and most of them have more than one family member. However, novel miRNAs are usually weakly

expressed as previously reported in the spleen of Chinese giant salamander [88] and common carp [19]. Most known miRNAs are conserved and they are often widespread. Five miRNAs, including miR-21, let-7a, miR-26a-5p, miR-451, and miR-142a-5p, were among the top 10 miRNAs with the highest abundance in the grass carp spleen in this study (Table 1) and in the spleen of the Chinese giant salamander in a previous report [88]. The above results indicate that the miRNA expression profile of spleen may have a high consistency among vertebrates.

Generally, abundant miRNAs play fundamental and broad regulatory functions in maintaining biological processes. Interestingly, many of the known miRNAs identified in this study were highly expressed where miRNAs with TPM > 100 reached more than 39%. In particular, ten, seven, five, four and four members in the family of let-7, mir-10, mir-30, mir-126 and mir-15, respectively, were expressed in high abundance in grass carp spleen (Table 1). Examination of the literature for miRNAs of TPM > 1000, it was found that these miRNAs were reported in other species and closely involved in animal immunity (Table 1). This included innate and adaptive immune responses, T-cell activation and cytotoxicity, innate anti-viral response, phagocytosis of myeloid inflammatory cells, early iNKT cell development, inflammatory factor secretion from macrophages, macrophage polarization, hypoxia-induced immunosuppression, Treg-mediated immunological tolerance, and inflammatory response to bacterial infection (Table 1). These suggest that the abundance miRNAs in the grass carp spleen may function similar to those of their orthologs and play important roles in regulating the spleen's physiological function.

The expression of miRNAs usually exhibits temporal and spatial specificity. In this study, twenty differentially expressed miRNAs between one-year-old and three-year-old grass carp spleens were identified and closely related to hematopoiesis and immunity as determined by functional enrichment analysis (Figs 3 and 4). Most of the differentially expressed miRNAs were down-regulated compared with the one-year-old spleen. Except for miR-192, miR-144-5p, miR-2188-5p, miR-451 and miR-144-3p, these miRNAs were expressed in relatively low abundance during grass carp spleen development (S3 Table). Interestingly, in these differentially expressed miRNAs, target gene prediction revealed potential interactions between miR-144-3p and miR-205-5p with *Tlx1*, miR-144-5p with *Bapx1*, and miR-192 with *Wt1*. It is known that *Tlx1*, *Bapx1* and *Wt1* are major marker genes for spleen development. In particular, transcription factor *Tlx1* is a marker for early spleen mesenchymal cells controlling cell fate specification and organ expansion during spleen development [99]. These studies suggest that differentially expressed miRNAs play important roles in regulating spleen immune function in grass carp.

A single miRNA can regulate multiple target genes or a single gene can be targeted by multiple miRNAs simultaneously. These miRNAs construct a complex network and collectively regulate gene expression. The spleen has important functions in haematopoiesis and immunity. The spleen's development is achieved through complex interactions between spleen mesenchymal cells and invading hematopoietic and endothelial cells, involving a series of cellular events. To reveal the regulatory mechanism of miRNAs underlying the spleen's physiological function in grass carp, miRNA–mRNA interaction networks related to haematopoiesis and immunity, and cell proliferation and differentiation were constructed (Figs 5 and 6). In these interaction networks, some genes, as a component of multiple pathways, link multiple pathways to each other. For instance, *NFKBIA* links across seven immune-related pathways such as toll-like receptor signaling pathway, cytosolic DNA-sensing pathway, B cell receptor signaling pathway, chemokine signaling pathway, RIG-I-like receptor signaling pathway, NOD-like receptor signaling pathway and T cell receptor signaling pathway. Likewise, *P38* links across five pathways closely related to cell proliferation and differentiation including TNF signaling pathway, MAPK signaling pathway, VEGF signaling pathway, Rap1 signaling pathway and

FoxO signaling pathway. Some miRNAs regulate the functions of multiple pathways by targeting multiple genes in these interaction networks. Thus, it is proposed that a complex regulatory network, formed by the interaction between miRNAs and their target gene and between pathways, regulates the spleen's physiological function in grass carp. Among the miRNAs constituting the above interaction network, some miRNAs including miR-144-3p, miR-192, and miR-100-3p have previously been demonstrated to be involved in immunity [42, 54, 69]. In particular, two members of the mir-144 family have potential target relationships with seven genes in three interaction networks, including *TLR5*, *IL11*, *NFKBIA*, *P38*, *SOCS3*, *ACSL*, and *HGF*. MiRNAs from the mir-144 family have been shown not only to participate in immunity, but also play an important role in hematopoietic regulation [100, 101]. Therefore, the interactive networks constructed in this study may truly reflect the complexity of the post-transcriptional regulation of the spleen's physiological function in grass carp.

## Supporting information

**S1 Fig. Post-sequencing analysis of sRNAs from the spleen of grass carp.** (A) Length distribution of clean reads. (B) Venn diagram of total sRNA between the cis1 and cis3. (C) Venn diagram of unique sRNA between the cis1 and cis3. (D) Annotation of unique sRNA in the RFam 11.0 database. (E) Number and distribution of clean reads mapped to the reference genome sequence.
(TIF)

**S2 Fig. MiRNA expression analysis in grass carp spleen.** (A) Histogram of miRNA expression distribution in cis1. (B) Histogram of miRNA expression distribution in cis3. (C) Boxplot of miRNA expression distribution in cis1 and cis3. (D) Scatterplot comparing the number of miRNA reads for cis1 and cis3. (E) Heatmap of the differentially expressed miRNAs between cis1 and cis3.
(TIF)

**S1 Table. The conserved miRNAs expressed in the spleen of grass carp.**
(DOCX)

**S2 Table. List of novel miRNAs identified in grass carp spleen.**
(DOCX)

**S3 Table. The significant differentially expressed miRNAs identified in this study.**
(DOCX)

## Acknowledgments

We thank Majorbio (Shanghai, China) for help with bioinformatic analysis and Dr. Xiaojun Liu (College of Animal Science and Technology, Henan Agricultural University) for proofreading the manuscript.

## Author Contributions

**Data curation:** Shengxin Fan.

**Funding acquisition:** Yinli Zhao.

**Project administration:** Guoxi Li.

**Validation:** Pengtao Yuan.

**Writing – original draft:** Yinli Zhao.

**Writing – review & editing:** Guoxi Li.

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
