## [Decision Letter · Decision Letter 0]

25 Jan 2022

PONE-D-21-39912Expression characteristics and interaction networks of microRNAs in spleen tissues of grass carp (Ctenopharyngodon idella)PLOS ONE

Dear Dr. Li,

Thank you for submitting your manuscript to PLOS ONE. After careful consideration, we feel that it has merit but does not fully meet PLOS ONE’s publication criteria as it currently stands. Therefore, we invite you to submit a revised version of the manuscript that addresses the points raised during the review process.

We look forward to receiving your revised manuscript.

Kind regards,

Academic Editor

PLOS ONE

Journal Requirements:

"This work was supported by grants from the Project of Henan University of Technology（CN）(No. 2017XTCX04 ). The funding bodies had no role in the study design or in any aspect of the collection,

604 analysis and interpretation of the data or in the writing of the manuscript." 

Additional Editor Comments:

All data and its representation including raw sequencing data (NCBI) should be accessible for the reviewers.

Reviewers' comments:

Reviewer's Responses to Questions

**Comments to the Author**

1. Is the manuscript technically sound, and do the data support the conclusions?

Reviewer #1: Yes

Reviewer #2: Partly

Reviewer #3: Yes

2. Has the statistical analysis been performed appropriately and rigorously? 

Reviewer #1: Yes

Reviewer #2: I Don't Know

Reviewer #3: Yes

3. Have the authors made all data underlying the findings in their manuscript fully available?

Reviewer #1: Yes

Reviewer #2: No

Reviewer #3: Yes

4. Is the manuscript presented in an intelligible fashion and written in standard English?

Reviewer #1: Yes

Reviewer #2: Yes

Reviewer #3: Yes

5. Review Comments to the Author

Reviewer #1: In their manuscript “Expression characteristics and interaction networks of microRNAs in spleen tissues of grass carp (Ctenopharyngodon idella)” Yinli et al. studied miRNA transcriptome and miRNA–mRNA interaction networks in grass carp normal spleen tissue. Specifically, they show:

• Identified 324 known conserved miRNAs and 9 novel miRNAs by using bioinformatics,

• 23 families of miRNAs showed highly conservation between vertebrates and invertebrates and 14 of them are present only in fish,

• Expression patterns of miRNAs consistent between one-year-old and three-year-old spleen.

This study is descriptive and provides resources for understanding the molecular details of the spleen development in the grass carp.

The presented conclusions are well supported by the data. Appropriate controls have been used to support the experiments.

Reviewer #2: In this manuscript Zhao et.al. report expression of sRNAs in the spleen of grass carp (Ctenopharyngodon idella). Using the sRNA expression data, the authors have identified conserved as well as unique microRNA in young and aged grass carp spleen and predicted potential targets and interaction networks. This report is potentially important considering the importance of microRNAs in the regulation of fish immunity and spleen function. However, the data are not fully accessible/legible in order to make evaluation of the manuscript.

1. Please rectify the raw data archive accession number. The information is submitted which says, “The raw sequencing data had been deposited into the NCBI database Sequence Read Archive under number PRJNA328412 (mRNA) and PRJNA645243 (miRNA)”. However, no entry was found using accession number PRJNA645243. Inaccessibility of the data may impede the conclusions of the paper.

2. Figures are illegible, specifically Figures 1,3,5 and 6 are inaccessible to make any assessment of the data presented. Other figures also need significant improvement in presentations.

Minor

1. Figure 4: Please provide the qPCR primer sequences or catalog number if commercially available.

Line 17 and 99: Six versus two library construction were respectively mentioned. Authors needs to clarify the differences.

Reviewer #3: The spleen is found in nearly all vertebrates and is one of the major lymphoid organs. It primarily acts as a blood filter thereby recognizing and removing old or damaged red blood cells, and also plays important role in regard to the immune system. Previously, microRNAs are known to play a vital role in regulating the immune function of the spleen, although their expression and interaction networks are still poorly understood in late post-natal development.

The study “Expression characteristics and interaction networks of microRNAs in spleen tissues of grass carp (Ctenophariyngodon idella)” by Zhao et al. reported for the first time the identification of the miRNA transcriptome and also provided the miRNA-mRNA interaction networks. They have constructed small RNA libraries from spleen tissues of one and three year old grass carp, and with the help of bioinformatic analysis have identified a total of 324 known conserved miRNAs and 9 novel miRNAs. The authors showed that 23 families were highly conserved between vertebrates and invertebrates while 14 others were present only in fish, and miRNA expression patterns were highly consistent between the two spleen groups. The data obtained have been expanded to the number of known miRNAs, thus providing important information to better understand spleen development of grass carp at the molecular level.

The idea and study designed is well conceived and well performed, and the findings in the manuscript are very interesting. The data and results are well presented and very convincing. Overall, the authors have done a wonderful job, however, there minor points that need to be addressed prior publication.

1. The main conclusion from the abstract and the introduction stresses upon the novel finding of the miRNA transcriptome and miRNA-mRNA interaction network, whereas in the discussion section the conclusion was stressed upon miRNA regulation of the lipid metabolism and spleen function. Kindly rearrange such that a common conclusion is presented throughout the manuscript.

2. In Line 99, it says “two miRNA libraries were constructed from the spleen tissue of one year old and three year old”, whereas in line 140 it says “ six miRNA libraries were constructed from spleen tissue of grass carp at one year and three years of age”. So what is the exact number of libraries being constructed?

3. The resolution of the figures need further enhancement, as it is difficult to visualize and read.

4. Although, it is mentioned that RNA isolation from spleen tissues was carried out using Trizol. But what type of method was used to homogenize the spleen tissues?

5. What is the amount of total RNA used to make each miRNA library construction by TruSeq Small RNA Sample prep Kit for sequencing purposes?

6. For qRT-PCR method section, kindly provide a brief description on how the cDNA was made? Also please include a reference of the method used for calculation of the miRNA expression level.

7. Kindly maintain a uniformity of the numbering for figures including legends and text in the results section. For instance, Figure 1A is presented differently in the figure legends as (a) instead of (A), and the same is for figure 3, 4, and 6.

8. The subfigure number A, B, C,… and F should be also mentioned in the legend for figure 4 along with the respective miRNA.

9. The figure legends for figures 7, 8 and 9, should come after the result description, but not before as presented in the manuscript.

10. Lines 380, 391 and 400, the sentences start with the same words “ In the miRNA-mRNA interaction networks associated with”, kindly reframe each of the sentences differently.

6. PLOS authors have the option to publish the peer review history of their article (what does this mean?). If published, this will include your full peer review and any attached files.

Reviewer #1: No

Reviewer #2: No

Reviewer #3: No

---

## [Author Response · Author response to Decision Letter 0]

18 Feb 2022

Dear editor and reviewers,

Thank you very much for your hard work on my manuscript. According to the review comments of editors and reviewers and the requirements of the magazine, we have made a thorough revision of the manuscript entitled “Expression characteristics and interaction networks of microRNAs in spleen tissues of grass carp (Ctenopharyngodon idella)” (PONE-D-21-39912). The sections marked in red are the modified contents in the “Revised Manuscript with Track Changes”.

We hope that our revised manuscript could be accepted for publication.

Sincerely,

Guoxi Li

The response to Journal Requirements

Point 1: Please ensure that your manuscript meets PLOS ONE's style requirements, including those for file naming.

Response 1: We have revised our manuscript completely according to PLOS ONE's style requirements, including figures, table, references, language,and etc.. See the revised text.

Point 2: We suggest you thoroughly copyedit your manuscript for language usage, spelling, and grammar.

Response 2: Our original manuscript has been sent to Asia Science Editing for the English language editing before the initial submission. The language of revised manuscript has been refined and polished by Dr. Xiaojun Liu ( Henan Agricultural University). Therefore, we believe that the language of our manuscript can meet the requirements of the magazine.

Point 3: We note that the grant information you provided in the ‘Funding Information’ and ‘Financial Disclosure’ sections do not match.

Response 3: I'm sorry, this is caused by the operation error. In fact, we have provided the correct grant numbers in the ‘Funding Information’ section. But we mismanaged in the ‘Financial Disclosure’ sections. We have amended "Funding Statement" within our cover letter as follows:

"This work was supported by grants from the Project of Henan University of Technology（CN）(No.2017XTCX04)."

Please help to change it online. Thank you!

Point 4: We note that you have provided funding information that is not currently declared in your Funding Statement. However, funding information appears in the Acknowledgments section.

Response 4: According to your comments, we have removed any funding-related text from the manuscript. We would like to update our Funding Statement as follows:

"This work was supported by grants from the Project of Henan University of Technology（CN）(No.2017XTCX04)."

Please help to change it online. Thank you!

Point 5: All data and its representation including raw sequencing data (NCBI) should be accessible for the reviewers.

Response 5: We have declared that all data are fully available without restriction in Data Availability. At the same time, we also state the following in the Data Availability Statement section of our manuscript:

"The raw sequencing data had been deposited into the NCBI database Sequence Read Archive under number PRJNA328412 (mRNA) and PRJNA645243 (miRNA). The other data and material used and analyzed in this study are within the manuscript and its Supporting Information files."

Therefore, all data for this study are available.

However, the Reviewer#2 mentioned that no entry was found using accession number PRJNA645243, and the main reason was the data release date setting problem. The data was originally set to be released on July 7, 2024. To this end, we have made corresponding adjustments. 

See: https://www.ncbi.nlm.nih.gov/bioproject/PRJNA645243

The response to comments from Reviewer #1

In their manuscript , Yinli et al. studied miRNA transcriptome and miRNA–mRNA interaction networks in grass carp normal spleen tissue. This study is descriptive and provides resources for understanding the molecular details of the spleen development in the grass carp. The presented conclusions are well supported by the data. Appropriate controls have been used to support the experiments. 

Response: Thank you very much for your objective evaluation of our manuscript.

The response to comments from Reviewer #2

Thank you for your objective evaluation of our manuscript. We have revised the manuscript in accordance with your comments.

Point 1: Please rectify the raw data archive accession number. The information is submitted which says, “The raw sequencing data had been deposited into the NCBI database Sequence Read Archive under number PRJNA328412 (mRNA) and PRJNA645243 (miRNA)”. However, no entry was found using accession number PRJNA645243. Inaccessibility of the data may impede the conclusions of the paper.

Response 1: I'm sorry. The main reason was the data release date setting problem. The data was originally set to be released on July 7, 2024. To this end, we have made corresponding adjustments.

See: https://www.ncbi.nlm.nih.gov/bioproject/PRJNA645243

Point 2: Figures are illegible, specifically Figures 1,3,5 and 6 are inaccessible to make any assessment of the data presented. Other figures also need significant improvement in presentations.

Response 2: I am very sorry for the trouble caused to your viewing. This may be caused by file compression when the system generates PDF. In fact, the original image downloaded from the submission system is quite clear.

According to your comments, we have improved all the figures, further increased the resolution and changed the presentations of some figures.

In addition, the figure 1and figure 3 in the original version of the manuscript have been used as supporting information to simplify and refine our manuscript.

Point 3: Figure 4: Please provide the qPCR primer sequences or catalog number if commercially available.

Response 3: The qRT-PCR analysis for validating six SDE miRNAs was performed using the kits customized at the RiboBio company (Guangzhou, China). The qPCR primer sequences were designed by the company based on the mature sequences and precursor sequence of the identified miRNAs. However, due to intellectual property rights, the company did not provide us with primers related information. Therefore, we are very sorry that we cannot provide these primer sequences in manuscript.

Point 4: Line 17 and 99: Six versus two library construction were respectively mentioned. Authors needs to clarify the differences.

Response 4: I'm very sorry, this is due to our spelling mistake. In fact, we constructed two libraries in this study. We have revised this error, see the line 19 in the revised text.

The response to comments from Reviewer #3

Thank you very much for your objective evaluation of our manuscript. We have revised the manuscript in accordance with your comments.

Point 1: The main conclusion from the abstract and the introduction stresses upon the novel finding of the miRNA transcriptome and miRNA-mRNA interaction network, whereas in the discussion section the conclusion was stressed upon miRNA regulation of the lipid metabolism and spleen function. Kindly rearrange such that a common conclusion is presented throughout the manuscript.

Response 1: Thank you for your comments. According to your opinion, we have deleted the content relating to lipid metabolism in the line 517-527 of paragraph 5, line 541-546 of paragraph 6, and the last paragraph in the discussion section of original manuscript. Thus, the main conclusion throughout the manuscript stresses upon the characteristics of the miRNA transcriptome profile and miRNA-mRNA interaction network in the spleen. At the same time, we also adjusted the corresponding references. See the revised text.

Point 2: In Line 99, it says “two miRNA libraries were constructed from the spleen tissue of one year old and three year old”, whereas in line 140 it says “ six miRNA libraries were constructed from spleen tissue of grass carp at one year and three years of age”. So what is the exact number of libraries being constructed?

Response 2: I'm very sorry, this is due to our spelling mistake. In fact, we constructed two libraries in this study. We have revised this error, see the line 168 in the revised text.

Point 3: The resolution of the figures need further enhancement, as it is difficult to visualize and read.

Response 3: I am very sorry for the trouble caused to your viewing and reading. This may be caused by file compression when the system generates PDF. In fact, the original image downloaded from the submission system is quite clear.

According to your comments, we have improved all the figures, further increased the resolution and changed the presentations of some figures.

In addition, the figure 1and figure 3 in the original version of the manuscript have been used as supporting information to simplify and refine our manuscript.

Point 4: Although, it is mentioned that RNA isolation from spleen tissues was carried out using Trizol. But what type of method was used to homogenize the spleen tissues?

Response 4: The spleen tissues were homogenized by grinding in a mortar. We have supplemented this method in the "Sample collection and RNA isolation" section, see the line 158-159 in the revised text.

Point 5: What is the amount of total RNA used to make each miRNA library construction by TruSeq Small RNA Sample prep Kit for sequencing purposes?

Response 5: The initial total RNA was 1μg for each miRNA library construction. We have supplemented this description in the "MiRNA library construction and post-sequencing analysis" section, see the line 171 in the revised text.

Point 6: For qRT-PCR method section, kindly provide a brief description on how the cDNA was made? Also please include a reference of the method used for calculation of the miRNA expression level.

Response 6: Thank you for your comments. According to your comments, we have supplemented a brief description of cDNA synthesis in the "Quantitative real-time PCR (qRT-PCR) analysis" section, see the line 249-253 in the revised text. At the same time, we also added a reference of the method used for calculation of the miRNA expression level, see the line 260 and line 786-788 in the revised text.

Point 7: Kindly maintain a uniformity of the numbering for figures including legends and text in the results section. For instance, Figure 1A is presented differently in the figure legends as (a) instead of (A), and the same is for figure 3, 4, and 6.

Response 7: We have revised them according to your comments. See the revised text.

Point 8: The subfigure number A, B, C,… and F should be also mentioned in the legend for figure 4 along with the respective miRNA.

Response 8: We have revised them according to your comments. See the line 363-366 in the revised text.

Point 9: The figure legends for figures 7, 8 and 9, should come after the result description, but not before as presented in the manuscript.

Response 9: Thank you so much for your comments. We have adjusted them according to your opinion. See the line 407-469 in the revised text.

Point 10: Lines 380, 391 and 400, the sentences start with the same words “ In the miRNA-mRNA interaction networks associated with”, kindly reframe each of the sentences differently.

Response 10: Thank you so much for your comments. We have revised these sentences according to your opinion. See the line 411, line 429 and line 452 in the revised text.

---

## [Decision Letter · Decision Letter 1]

16 Mar 2022

Expression characteristics and interaction networks of microRNAs in spleen tissues of grass carp (Ctenopharyngodon idella)

PONE-D-21-39912R1

Dear Dr. Li,

We’re pleased to inform you that your manuscript has been judged scientifically suitable for publication and will be formally accepted for publication once it meets all outstanding technical requirements.

Kind regards,

Academic Editor

PLOS ONE

Additional Editor Comments (optional):

Reviewers' comments:

Reviewer's Responses to Questions

**Comments to the Author**

1. If the authors have adequately addressed your comments raised in a previous round of review and you feel that this manuscript is now acceptable for publication, you may indicate that here to bypass the “Comments to the Author” section, enter your conflict of interest statement in the “Confidential to Editor” section, and submit your "Accept" recommendation.

Reviewer #1: All comments have been addressed

Reviewer #2: All comments have been addressed

Reviewer #3: All comments have been addressed

2. Is the manuscript technically sound, and do the data support the conclusions?

Reviewer #1: Yes

Reviewer #2: Yes

Reviewer #3: Yes

3. Has the statistical analysis been performed appropriately and rigorously? 

Reviewer #1: Yes

Reviewer #2: Yes

Reviewer #3: Yes

4. Have the authors made all data underlying the findings in their manuscript fully available?

Reviewer #1: Yes

Reviewer #2: Yes

Reviewer #3: Yes

5. Is the manuscript presented in an intelligible fashion and written in standard English?

Reviewer #1: Yes

Reviewer #2: Yes

Reviewer #3: Yes

6. Review Comments to the Author

Reviewer #1: For Fig 3 and 4, even though authors did triplicate, but didn't show error bars. Please check and correct it.

Reviewer #2: Authors have addressed all my comments, made data and figures legible. The data support the conclusions of the manuscript.

Reviewer #3: Plos One Reports-PONE-D-21-39912R1:

In their manuscript “Expression characteristics and interaction networks of microRNAs in spleen tissues of grass carp (Ctenophariyngodon idella)”, Zhao et al. studied on miRNA transcriptome and the miRNA-mRNA interaction networks. This detailed study and data obtained do provides important resources at the molecular levels in understanding how spleen develops in grass carp.

The authors have done a wonderful job in accommodating all the comments that were suggested and I am very pleased to recommend the present version of the manuscript for publication in this journal.

7. PLOS authors have the option to publish the peer review history of their article (what does this mean?). If published, this will include your full peer review and any attached files.

Reviewer #1: No

Reviewer #2: No

Reviewer #3: No

---

## [Editor Report · Acceptance letter]

18 Mar 2022

PONE-D-21-39912R1 

Expression characteristics and interaction networks of microRNAs in spleen tissues of grass carp (*Ctenopharyngodon idella*) 

Dear Dr. Li:

I'm pleased to inform you that your manuscript has been deemed suitable for publication in PLOS ONE. Congratulations! Your manuscript is now with our production department. 

Kind regards, 

on behalf of

Dr. Rajakumar Anbazhagan 

Academic Editor

PLOS ONE